# Metabolic Syndrome in the Amazon: Customizing Diagnostic Methods for Urban Communities

**DOI:** 10.3390/nu17030538

**Published:** 2025-01-31

**Authors:** José M. Alcaide-Leyva, Manuel Romero-Saldaña, María García-Rodríguez, Rafael Molina-Luque, María del Rocío Jiménez-Mérida, Guillermo Molina-Recio

**Affiliations:** 1Department of Nursing, Pharmacology and Physiotherapy, Faculty of Medicine and Nursing, University of Cordoba, 14014 Cordoba, Spain; jmalcaide@uco.es (J.M.A.-L.); p72molur@uco.es (R.M.-L.); en1moreg@uco.es (G.M.-R.); 2Associated Research Group GA16 Lifestyles, Technology and Health, Maimonides Institute for Biomedical Research of Cordoba (IMIBIC), 14014 Cordoba, Spain; 3Department of Nursing and Nutrition, Faculty of Biomedical and Health Sciences, Universidad Europea de Madrid, Calle Tajo S/N, 28670 Madrid, Spain

**Keywords:** metabolic syndrome, diagnostic model, urban Amazonian population, clinical decision tree, public health, early detection

## Abstract

**Background/Objectives**: Metabolic syndrome is a significant public health issue, particularly in urbanizing regions like the Peruvian Amazon, where lifestyle changes have increased the prevalence of metabolic disorders. This study aimed to develop and validate a simple, cost-effective diagnostic model for early detection of metabolic syndrome in the urban population of San Juan Bautista, Iquitos. **Methods**: A cross-sectional study was conducted with 251 adults aged over 18 years. Data collection included anthropometric measurements, body composition analysis, and biochemical assessments. Logistic regression analyses identified key predictors of metabolic syndrome, and clinical decision trees were developed to enhance diagnostic accuracy. **Results**: The prevalence of metabolic syndrome was 47.9%. Systolic blood pressure, triglycerides, and very-low-density lipoprotein cholesterol were the strongest predictors. The most effective diagnostic model, combining very-low-density lipoprotein cholesterol and systolic blood pressure, achieved a sensitivity of 91.6% and a specificity of 78.5%, demonstrating high diagnostic accuracy. **Conclusions**: The proposed model offers a practical, low-cost tool for early detection of metabolic syndrome in resource-limited urban settings. However, its findings are limited by the small sample size and the lack of external validation, requiring further studies to confirm its generalizability and applicability to other populations. Its implementation in primary healthcare could facilitate timely interventions, reducing the risk of chronic diseases in vulnerable populations.

## 1. Introduction

### 1.1. Metabolic Syndrome: Definition and Global Relevance

Metabolic syndrome (MetS) is a set of metabolic disturbances that significantly increase the risk of developing chronic non-communicable diseases (NCDs), such as type 2 diabetes mellitus (T2D) and cardiovascular disease (CVD). The main criteria defining MetS include abdominal obesity, hyperglycaemia, high blood pressure (HBP), hypertriglyceridaemia and reduced high-density lipoprotein (HDL) levels [1,2]. This set of interrelated risk factors significantly increases the likelihood of severe complications, doubling the risk of cardiovascular events and increasing the risk of developing T2D by fivefold [3,4]. Globally, MetS affects approximately 25% of the adult population, and its prevalence has been increasing in recent decades due to various environmental and lifestyle factors, especially in urban settings where diet and physical activity levels have changed dramatically [5,6].

Accelerated urbanisation and the Nutritional Transition are among the main factors contributing to the increase of METS. In many regions of the world, dietary habits have shifted towards increased consumption of ultra-processed and high-calorie foods rich in saturated fats, sugars and salt, while consumption of fresh and nutritious foods has declined [7]. At the same time, changing lifestyles have reduced physical activity levels due to the increasing adoption of sedentary jobs and the use of technologies that facilitate transport and reduce the need for physical exertion. This context has significantly increased the prevalence of obesity and other risk factors contributing to MetS in urban populations from diverse cultural and economic backgrounds [8].

### 1.2. Prevalence and Risk Factors for MetS in Peru and Latin America

In Latin America, MetS is an increasingly relevant public health problem, with a steadily growing prevalence. Factors such as urbanisation, an increasing per capita income and the globalisation of dietary patterns have transformed the diet and lifestyle of Latin American populations, placing them at high risk of metabolic diseases [9]. In Peru, the prevalence of MetS reaches 31% in adults, reflecting the severity of this problem nationwide. Moreover, prevalence is particularly high in urban areas compared to rural areas, indicating a direct relationship between urbanisation and the increase in risk factors associated with MetS, such as overweight, abdominal obesity and dyslipidaemia [10]. This phenomenon is attributed to changes in lifestyle habits, with increasing access to ultra-processed foods and a decrease in physical activity levels that characterises urban areas of the country [11,12].

### 1.3. MetS in the Peruvian Amazon and the Impact of Urbanisation

The Peruvian Amazon faces unique public health challenges, and MetS is one of the emerging problems in this region. The ‘double burden’ of malnutrition is a distinctive feature of the Peruvian Amazon, where undernutrition and an increased prevalence of overweight and obesity coexist. This complex situation is driven by a nutritional transition in a region that has historically been dependent on local, nutrient-rich foods, but which in recent decades has experienced an increase in the availability of processed and high-calorie foods. This dietary change, coupled with increasingly sedentary urban lifestyles, has increased the burden of chronic non-communicable diseases, of which MetS is central [13,14].

Iquitos, the largest city in the Peruvian Amazon and capital of the department of Loreto, presents a particular context in this sense. Being unconnected by road to the rest of the country and dependent on river and air routes, Iquitos faces a series of limitations that complicate access to health services and to a varied and high-quality nutritional diet. These conditions exacerbate the vulnerability of the Iquitos population to MetS and other metabolic diseases, as dietary options are often limited and geographic barriers hinder the implementation of effective public health policies [15].

### 1.4. San Juan Bautista: Characterisation of an Urban Amazonian District

Within Iquitos, the district of San Juan Bautista is one of the areas that best represents the challenges of urbanisation in the Peruvian Amazon. With a mainly urban population, San Juan Bautista has witnessed a change in the lifestyle and dietary habits of its inhabitants, which has significantly increased the risk factors for MetS in this community. Data indicate that abdominal obesity and dyslipidaemia are increasingly common in the district, affecting even individuals with a body mass index (BMI) considered normal by international standards [16,17]. These findings highlight the need to develop tailored diagnostic models that take into account the anthropometric and metabolic particularities of the Peruvian Amazonian population, where risk thresholds established for other populations may not adequately reflect the risk profile of this population [10,18].

### 1.5. The Harmonised MetS Diagnostic Model and Its Application in Specific Populations

To respond to variations in risk profiles between different ethnicities and regions, the harmonised model for the diagnosis of MetS, promoted by the International Diabetes Federation (IDF), has become an international standard for the identification of at-risk individuals. This model establishes specific cut-off points for key risk factors, such as waist circumference, adapting these values to reflect variations in body composition and metabolic susceptibility among different groups [19]. However, studies have highlighted that the application of this model in Latin American and Amazonian populations requires local adjustments, as these communities have particular susceptibility to MetS and its comorbidities, which may not be adequately captured in the standard criteria of the harmonised model [20,21]. Adapting international models to local contexts presents several challenges. First, the cut-off points for anthropometric and metabolic variables often do not reflect the unique characteristics of populations in regions such as the Peruvian Amazon, where body composition and genetic predispositions differ significantly from those in Western populations [20,22]. Second, cultural and socioeconomic factors, such as dietary patterns, access to healthcare, and levels of physical activity, vary greatly and influence the prevalence and risk factors of MetS [17,23]. Finally, the availability of resources and diagnostic tools in low-resource settings can limit the feasibility of implementing standardised criteria [13]. For instance, while the IDF model provides a global framework, its practical application in vulnerable communities like San Juan Bautista requires tailoring to account for these local realities. These challenges underscore the need for models that are both scientifically robust and contextually appropriate

In the case of the Peruvian Amazon, previous studies have shown that variables such as waist-to-height ratio (WHtR) and waist circumference (WC) are useful diagnostic tools in low-resource settings, as they allow for practical and accessible risk assessment without the need for advanced technology [24]. These anthropometric measures, adapted to specific cut-off points for the Amazonian population, facilitate early and affordable identification of estimated risk in urban settings such as San Juan Bautista, which is particularly important in areas where resources for diagnosis and treatment are limited.

Additionally, several diagnostic models validated in comparable populations provide further evidence supporting the use of tailored approaches in resource-limited contexts. For instance, the FINDRISC model has been shown to effectively predict cardiometabolic risk in low-resource environments through non-invasive measures [25]. Similarly, the Triglycerides and Glucose (TyG) index has demonstrated high diagnostic accuracy in Latin American populations by integrating simple biochemical parameters [26,27]. These models highlight the importance of adapting diagnostic tools to specific sociodemographic and cultural contexts, as they account for regional variations in metabolic and anthropometric profiles. Building upon this evidence, our study aims to develop a diagnostic model tailored specifically to the urban Amazonian population, addressing their unique characteristics and public health challenges.

## 2. Materials and Methods

The study population consisted of EsSalud (Social Health Insurance) workers at the health post in the district of San Juan Bautista in the city of Iquitos and the service users themselves. 

Study sample

A non-probabilistic convenience sample was used in which an attempt was made to include as many subjects as possible based on the time and technical resources available. In the end, 251 subjects were included in the study.

### 2.1. Eligibility Criteria

Persons over 18 years of age who agreed to participate in the study and sign the informed consent form were included, and those with an inability to stand upright during anthropometry and bioimpedance were excluded. 

○Study variables and measurementThe dependent variable of the study was the diagnosis of MetS according to the NCEP ATP III criteria [28]. These criteria establish the presence of STEM when three or more of the following risk factors are present: Obesity Central. CC ≥ 94 cm for males o ≥88 cm for females.
High blood pressure. SBP ≥ 130 mmHg and/or DBP ≥ 85 mmHg or antihypertensive treatment.High triglycerides. TG ≥ 150 mg/dL or lipid-lowering treatment.High blood glucose. FG ≥ 100 mg/dL or hypoglycaemic treatment.Low HDL. HDL < 40 mg/dL in women or HDL < 50 mg/dL in men or pharmacological treatment to address it.○The independent variables of the study were:
Sociodemographics: Age (years) and sex (male/female).Anthropometric. Height (cm), weight (kg), BMI (kg/m^2^), waist circumference (WC, cm), body fat percentage (FP%), muscle mass (MM, kg), basal metabolic rate (BMR, kcal), waist-to-height ratio (WHtR), A New Body Shape Index (ABSI) (113) and Body Adiposity Index (BAI) (114). In addition, the BMI was used to assess nutritional status according to the cut-off points established by the WHO (111) for underweight (≤18.49), normal weight (18.5–24.99 kg/m^2^), overweight (25.00–29.99 kg/m^2^) and obesity (≥30.00 kg/m^2^). WHtR was categorised as healthy (Males (M): 0.43 to 0.52 and Females (F): 0.42 to 0.48); overweight (M: 0.53 to 0.57 and F: 0.49 to 0.53); elevated overweight (M: 0.58 to 0.62 and F: 0.54 to 0.57), and obese (M: ≥0.63 and F: ≥0.58) [29,30].Laboratory tests. FG (mg/dL), HDL cholesterol (mg/dL), TG (mg/dL), VLDL cholesterol according to Friedewald’s formula (mg/dL) [31].

### 2.2. Data Collection

For data collection, several teams were formed, composed of nurses, nutritionists and students previously trained in data collection. The students belonged to the final courses of Nursing, Nutrition and Medicine at the University of the Peruvian Amazon. Regarding measurements, height was taken using a Seca 213 portable stadiometer (Seca. Hamburg, Germany). Body composition and weight were measured with a Tanita BC-545N (Tanita Corp., Itabashini-Ku, Tokyo, Japan). Bioimpedance analysis was chosen for assessing body composition due to its practicality, portability, and cost-effectiveness in resource-limited settings such as San Juan Bautista. While BIA has limitations compared to gold-standard methods like dual-energy X-ray absorptiometry, it is widely validated in population-based studies. BIA provides reliable estimates of body fat percentage, muscle mass, and basal metabolic rate, making it a suitable and effective tool for large-scale community-based studies. CC was measured at the midpoint between the lower limit of the last rib and the iliac crest. Both variables were collected with a Lufki W606PM (Lufki, Missouri City, TX, USA) metal tape with an accuracy of 0.1 cm (118). Both circumferences were measured at the end of a regular exhalation in an upright position with arms suspended alongside the torso [32]. Blood pressure was measured using an OMRON M4 digital sphygmomanometer (OMRON Corporation Ltd., Tokyo, Japan) according to blood pressure measurement standards. Finally, biochemical variables were measured with a Cardiocheck (pts Diagnostics, Whitestown, IN, USA) and PTS Panels self-testing strips (pts Diagnostics, USA) [33].

### 2.3. Statistical Analysis

Quantitative variables were presented using the mean and standard deviation. Qualitative variables were expressed as absolute frequency and percentages.

To study the goodness of fit of the quantitative variables to the normal distribution, the Kolmogorov–Smirnov test with the Lilliefors correction was used, together with the analysis of their histograms and Q-Q and P-P plots.

For the comparison of means, Student’s *t*-test or one-factor ANOVA was used when the parametricity criteria were met. In the latter case, post hoc contrasts were used using the Bonferroni test. In the absence of normality of the data, non-parametric tests (Mann–Whitney U and Kruskal–Wallis) were used. For hypothesis testing of qualitative variables, the chi-square test and Fisher’s exact test were used.

On the other hand, binary logistic regressions were performed. In this analysis, crude and adjusted Odds Ratios (OR) were obtained. The Wald test was applied as a statistical contrast method to assess the significance of the coefficients in the logistic regression model. The goodness-of-fit of the model was assessed using the Hosmer–Lemeshow test. Finally, to assess the predictive ability of the model, we calculated the Cox–Snell and Nagelkerke tests, together with the coefficients of variance and determination.

### 2.4. Ethical Considerations

Participants were treated within the bioethical legislative framework of the Republic of Peru. The guidelines of the Declaration of Helsinki [34], which establishes the fundamental ethical principles for medical research, were strictly followed. All participants were informed personally, verbally and in writing, of the objectives of the research study. The researchers also informed them of the dangers and advantages of their participation in this project. All informed consents were signed and retained.

To ensure inclusivity and respect for the cultural and social characteristics of the local population, the study involved collaboration with local healthcare professionals, including physicians, nutritionists, and nurses, who were familiar with the community and its sociocultural dynamics. Their participation facilitated trust-building and effective communication with participants. Additionally, recruitment materials and consent forms were adapted to the local context, using culturally appropriate language and formats to ensure comprehension and meaningful participation. Efforts were also made to achieve gender balance and include participants across a wide age range to reflect the diversity of the population.

In addition, the provisions of Law No. 26842—Peru’s General Health Law [35], which establishes the guiding principles of the health system in the country and addresses issues related to medical ethics and health research, were complied with. In the area of data protection, the regulations of the European Union’s General Data Protection Regulation (GDPR) [36] were applied, guaranteeing the privacy and rights of participants. This ethical and legal approach, both at local and European levels, ensured the protection of the rights and welfare of the study participants. The research project of this Doctoral Thesis was approved by the Research Ethics Committee of Cordoba (act 348/reference 5610).

## 3. Results

### 3.1. Characteristics of the Sample

The sample consisted of 251 participants, whose sociodemographic and anthropometric characteristics are shown in Table 1.

The mean age of the participants was 47.93 ± 15.71 years, with a higher proportion of women. In terms of anthropometric characteristics, the mean WC was 99.64 ± 9.65 cm, which determined that the majority of participants had a WC considered high. Similarly, the mean WhtR was 0.64 ± 0.067, also above the values indicating good nutritional status, which meant that 97.2% of the sample had a high WhtR. Finally, BMI averaged 30.36 ± 4.86 kg/m^2^, resulting in a high percentage of overweight and obese individuals.

Regarding the characteristics related to the data obtained from the bioimpedance and clinical variables (Table 2), a mean FP of 32.73 ±7.17% stood out, with 56.6% of the participants in the high level. Regarding the clinical variables, 80.5% of the participants had a high HDL-C and 41.4% a high TG. The mean AG was 103.42 ±41.79 mg/dL.

### 3.2. Bivariate Analysis and Logistic Regression for MetS

Data concerning bivariate analysis and logistic regression for MetS are shown in Table 3. In relation to the presence of MetS, the diseased population showed a mean age seven years older than the healthy. In addition, men had a much higher prevalence than women, being 5.54 times more likely to develop this syndrome in the former group.

Regarding anthropometric variables, WC, weight and height were associated with a higher prevalence of MetS (*p* < 0.001). However, WHtR did not show this relationship. The FP was significantly related to MetS (crude OR = 1.22, 95%CI: 1.05–1.43, *p* < 0.05). Similarly, other bioimpedance parameters such as MM, BMR and MA were related to the presence of this condition. This association was also evident for all clinical variables. Finally, the adjusted model included SBP, DBP, TG, FG and VLDL.

**Table 3 nutrients-17-00538-t003:** Bivariate analysis and crude and adjusted logistic regression for MS.

Variable	No MS 84 (33.5%)	Yes MS 167 (66.5%)	Raw OR	*p*	Adjusted OR	*p*
**Age (years)**	43.74 (14.4)	50.04 (14.51)	1.03 (1.01–1.05)	*p* < 0.05		NS
**Sex**						
**Women**	73 (86.9%)	91 (54.5%)	1			
**Men**	11 (13.1%)	76 (45.5%)	5.54 (2.74–11.19)	*p* < 0.001		NS
**Anthropometry**						
**WC (cm)**	96.31 (8.55)	101.31 (9.76)	1.06 (1.02–1.09)	*p* < 0.001		NS
**Level of WC**						
**Low**	5 (6%)	2 (1.2%)	1			
**High**	79 (94%)	165 (98.8%)	5.22 (0.99–27.5)	*p* = 0.51		NS
**Weight (kg)**	86.91 (11.75)	76.29(16.1)	1.03 (1.01–1.05)	*p* < 0.001		NS
**Height (cm)**	152 (6.2)	155 (8.6)	1.08 (1.02–1.14)	*p* < 0.001		NS
**WHtR**	0.63 (0.05)	0.64 (0.07)	1.24 (0.77–2.01)	NS		
**Level of WHtR**						
**Low**	3 (3.6%)	2 (1.2%)	1			
**High**	81 (96.4%)	163 (98.8%)	3.01 (0.49– 18.42)	NS		
**BMI (kg/m^2^)**	29.45 (4.19)	30.81 (5.12)	1.06 (1.001–1.12)	*p* < 0.05		NS
**Level of BMI**						
**Underweight**	1 (1.3%)	1 (0.6%)				NS
**Healthy weight**	8 (10%)	15 (9.4%)				
**Overweight**	40 (50%)	56 (35%)				
**Obesity**	31 (38.8%)	88 (55%)				
**Variable**	MetS No 84 (33.5%)	MetS Yes 167 (66.5%)	Raw OR	*p*	Adjusted OR	*p*
**Bioimpedance variables**						
**FP (%)**	34.97 (7.94)	31.61 (6.489)	1.22 (1.05–1.43)	*p* < 0.05		NS
**MM (kg)**	42.34 (6.96)	49.03 (10.38)	1.09 (1.05–1.12)	*p* < 0.001		NS
**BM (kcal)**	1364 (189.73)	1556.31 (319.31)	1.003 (1.002–1.004)	*p* < 0.001		NS
**MA (years)**	46.61 (10.18)	53.19 (10.63)	1.06 (1.03–1.09)	*p* < 0.001		NS
**Clinical variables**						
**SBP (mmHg)**	115.27 (17)	137.96 (22.91)	1.06 (1.04–108)	*p* < 0.001	1.08 (1.05–1.11)	*p* < 0.001
**DBP (mmHg)**	72.61 (8.66)	82.3 (13.72)	1.07 (1.04–1.11)	*p* < 0.001	1.23 (1.12–1.28)	*p* < 0.001
**HDL-C: (mg/dL)**	43.96 (12.54)	35.26 (11.15)	0.91 (0.89–0.95)	*p* < 0.001		NS
**Level of HDL-C**						
**Low**	52 (61.9%)	150 (89.8%)	1			
**High**	32 (38.1%)	17 (10.2%)	0.18 (0.09–0.35)	*p* < 0.001		NS
**TG (mg/dL)**	98.7 (31.02)	170.75 (71.04)	1.03 (1.02–1.04)	*p* < 0.001	1.03 (1.02–1.04)	*p* < 0.001
**TG level**						
**Low**	82 (97.6%)	65 (38.9%)	1			
**High**	2 (2.4%)	102 (61.1%)	64.38 (15.29–270.64)	*p* < 0.001		NS
**FG (mg/dL)**	90.29 (19.74)	110.02 (48)	1.03 (1.01–1.05)	*p* < 0.001	1.02 (1.01–1.09)	*p* < 0.001
**Level of FG**						
**Low**	80 (95.2%)	89 (53.3%)	1			
**High**	4 (4.8%)	78 (46.7%)	17.52 (6.13–50.04)	*p* < 0.001		NS
**VLDL**	45.48 (14.29)	82.48 (39.87)	1.06 (1.043–1.08)	*p* < 0.001	1.04 (1.01–1.07)	*p* < 0.001

Quantitative variables with mean and SD. Qualitative variables with absolute frequency and percentage; BMI: Body Mass Index; WC: Waist Circumference; FP: fat percentage; MM: muscle mass; BM: basal metabolism; MA: metabolic age; SBP: systolic blood pressure; DBP: diastolic blood pressure; HDL-C: cholesterol; TG: triglycerides; FG: fasting glucose; VLDL: very low-density lipoprotein. NS: No Significant.

### 3.3. Development of the Clinical Decision Tree

Figure 1 shows the ROC curves for all variables included in the regression model fitted for MetS detection. Thus, the cut-off points that showed the best Youden index are shown (Table 4).

The variables were grouped logically by selecting those with the least complexity of application in the populations of the area studied. Three diagnostic models were then developed based on the cut-off points obtained. For this purpose, the SBP and DBP variables were transformed into dichotomous categorical variables that were used to develop the tree without forcing the first variable (Figure 2) and thus test the effectiveness of diagnosing MetS based on non-invasive clinical variables. In the second tree (Figure 3), the level of TG was categorised into a dichotomous variable. Once transformed, and together with the dichotomous SBP, a mixed model was designed with invasive (but easy to measure with portable devices) and non-invasive variables, without forcing the first variable. Finally, for the third model (Figure 4), the VLDL variable was dichotomised from the cut-off point calculated using the Youden index. In this case, the first discriminant variable was forced to be VLDL and, using the CHAID system, DBP was discarded and SBP was finally selected (Table 5).

### 3.4. Comparison of the Diagnostic Efficacy of Clinical Decision Trees

As a final step, a comparative analysis of the three diagnostic trees (Table 5) was performed based on their diagnostic accuracy. Model 2 showed the highest sensitivity, with 92.22%, followed by Model 3, while Model 1 registered a slightly lower diagnostic accuracy.

On the other hand, specificity was highest for Model 3, with a value of 78.57%, followed by Model 1 and Model 2. AsimiMetSo, Model 3 also stood out for its Validity Index, with a value of 87.25%, nine points higher than Model 1, which had the lowest values.

In terms of Predictive Values, the PPV of Model 3 stood out with 89.47%. Model 2 showed a PPV of 82.80%, while Model 1 scored the lowest. However, the best Negative Predictive Value was evidenced by Model 2, with a value of 80%, followed, in that order, by Models 3 and 1,

Overall, and given that it was comparatively better than the rest on the basis of diagnostic accuracy metrics and, in particular, the Youden index, Model 3 was assessed as the most suitable for screening for STEMI in this type of population.

## 4. Discussion

The results obtained in this study reveal a high prevalence of MetS in the urban population of San Juan Bautista, reaching 47.9%, a figure considerably higher than that reported in other regions of Peru, where the average prevalence is around 31% (Ramírez et al., 2022), and well above the 19.7% found in rural Peruvian populations located below 1,000 m above sea level [37]. This finding highlights the severity of the problem and underscores the influence of changes in dietary habits and decreased physical activity in urban settings. The high proportion of abdominal obesity (66.5%) and dyslipidaemia (41.4%) observed in our sample aligns with the literature linking these factors to the development of MetS. Recent studies have confirmed that accelerated urbanisation in developing countries has intensified the risk factors associated with MetS, increasing its prevalence to 52% in certain urban areas [38,39].

The prevalence of MetS in this urban population is higher than that found in studies conducted in rural or mixed communities, which is consistent with research highlighting the impact of urbanisation and nutritional transition on the rise of metabolic diseases [40]. Furthermore, the proposed diagnostic model, based on simple variables such as SBP and VLDL levels, has proven to be effective and accessible, achieving a sensitivity of 91.6% and a specificity of 78.5%. These results are comparable to those reported by Fornari Laurindo et al. [41], who achieved a sensitivity of 89.4% and specificity of 75.2% using similar methods. Another study [42] also emphasized the need to adjust metabolic indicator cut-off points for different populations, supporting the relevance of adapted models like the one developed in this study.

The development of simple, cost-effective, and locally adapted diagnostic tools represents a significant advancement for the early detection of MetS in vulnerable urban populations. Implementing these models in primary healthcare centres could enhance the identification of at-risk individuals and facilitate the application of preventive interventions. For instance, the Triglyceride and Glucose (TyG) Index has emerged as a cost-effective diagnostic tool for various medical conditions, reflecting underlying insulin resistance, a key factor in many metabolic disorders [43]. Additionally, non-invasive methods utilizing machine learning models have been proposed for early and low-cost identification of MetS, demonstrating high sensitivity and convenience for large-scale screening [44]). Guzmán et al. [45] demonstrated that community interventions based on early detection of MetS can reduce the incidence of cardiovascular diseases by 25% and type 2 diabetes by 30%.

One of the main strengths of this study lies in the adaptation of the diagnostic model to the specific characteristics of the studied urban population. The use of easily obtainable and low-cost clinical variables increases the feasibility of its application in resource-limited settings. Additionally, the model shows an adequate balance between sensitivity and specificity. Recent studies have validated similar approaches, highlighting the importance of practical and adaptable models for MetS prevention [46].

Among the limitations, the relatively small sample size (n = 251) stands out, which could affect the generalizability of the results. Furthermore, the use of a non-probabilistic convenience sample and the focus on a specific urban Amazonian district may limit the applicability of these findings to broader populations. Additionally, this study focused primarily on the pathological and metabolic profile of the urban Amazonian population, without including variables such as dietary intake, physical activity levels, and stress management or alcohol consumption. These variables are critical in understanding the multifactorial nature of metabolic syndrome and its risk factors. Future studies should address these limitations by recruiting larger, more representative samples and incorporating lifestyle and behavioural data to provide a more comprehensive assessment of metabolic health. Furthermore, external validation of the diagnostic model in diverse demographic and geographic contexts is essential to confirm its generalizability and clinical applicability. Additionally, the model was validated in the same population in which it was developed, making it necessary to test its effectiveness in other urban populations. Likewise, the use of standard formulas to estimate certain biochemical markers may not accurately reflect the reality of this community. Recent research suggests the incorporation of emerging biomarkers to improve diagnostic accuracy [47]. Furthermore, the reliance on traditional anthropometric and biochemical variables in this study, while practical for resource-limited settings, may not capture the full spectrum of factors contributing to MetS. Advanced biomarkers, such as inflammatory markers or metabolomics, could provide deeper insights into the pathophysiology of MetS and enhance diagnostic precision. Additionally, the integration of digital health solutions, including wearable devices and mobile applications, offers opportunities for real-time monitoring, early detection, and personalised management of MetS. Future studies should explore these innovations to complement traditional diagnostic models and improve health outcomes in vulnerable populations [48].

Future research should focus on validating the proposed model in other urban populations with different sociodemographic characteristics. Moreover, it would be advisable to expand the sample size and consider the use of additional biomarkers to enhance diagnostic precision. The implementation of longitudinal studies would allow for the evaluation of the model’s predictive capacity over the long term and its impact on the prevention of chronic diseases. These would also provide a deeper understanding of how risk factors evolve over time and how early detection through the proposed model could influence the natural history of MetS. In addition, tracking individual trajectories could help assess the model’s capacity to predict long-term health outcomes, such as the onset of type 2 diabetes and cardiovascular diseases.Therefore, these research efforts are crucial for validating the model’s utility beyond its current context and ensuring its effective applicability across diverse populations. Additionally, validating the model in urban populations with varying sociodemographic and cultural characteristics would allow for the identification of potential modifications needed to optimise its applicability and effectiveness in different settings. Integrating digital health technologies, such as wearable devices and mobile applications, could facilitate continuous monitoring and personalised management of individuals at risk of developing MetS [49]. Additionally, exploring the role of gut microbiota and its modulation may offer new therapeutic avenues for MetS management [50].

## 5. Conclusions

This study highlights the alarming prevalence of MetS in the urban population of San Juan Bautista, emphasizing the urgent need for targeted public health interventions. The diagnostic model developed, based on simple and cost-effective clinical variables such as systolic blood pressure and VLDL levels, demonstrated high diagnostic accuracy with a sensitivity of 91.6% and specificity of 78.5%. While the model shows promise as a practical and adaptable tool for early detection and prevention strategies in resource-limited urban settings, its findings should be interpreted within the context of the study’s limitations. These include the cross-sectional design, which prevents causal inferences, the non-probabilistic sampling method, which limits generalizability, and the need for validation in other populations and geographic contexts. Future research should address these limitations to further refine and validate the model, ensuring its broader applicability and effectiveness in diverse settings.

Implementing such cost-effective diagnostic tools in primary healthcare services can significantly improve the early identification of individuals at risk, enabling timely interventions that could reduce the incidence of cardiovascular diseases and type 2 diabetes. However, to enhance its applicability and reliability, future studies should validate this model in diverse urban populations and incorporate emerging biomarkers for better diagnostic precision.

In conclusion, our findings underscore the necessity for proactive health policies that incorporate accessible diagnostic methods and preventive measures tailored to vulnerable urban populations. The integration of digital health technologies could further strengthen the monitoring and management of MetS, fostering more effective and sustainable public health outcomes. 

## Figures and Tables

**Figure 1 nutrients-17-00538-f001:**
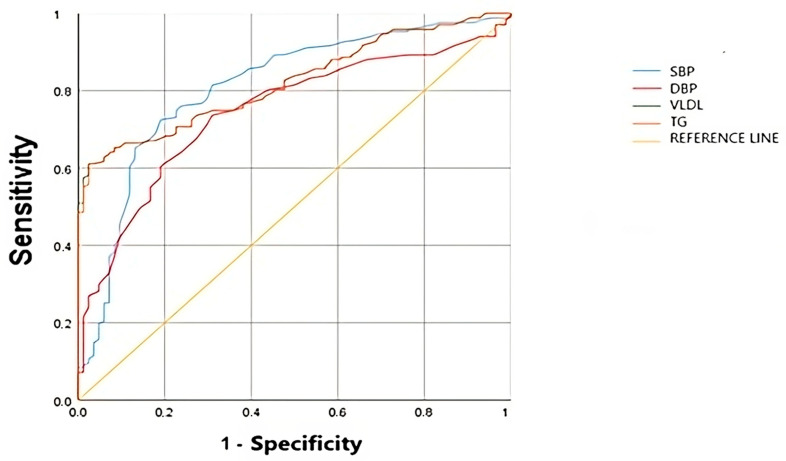
Area under the curve of the selected variables.

**Figure 2 nutrients-17-00538-f002:**
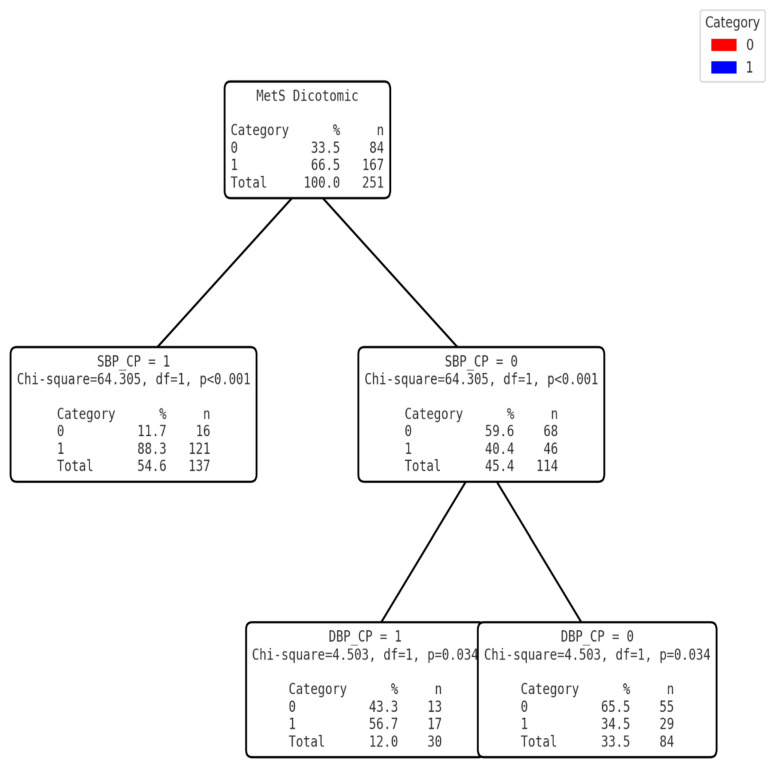
Clinical decision tree for MS based on SBP CP (SBP Cut-off point) and DBP CP (DBP Cut-off point).

**Figure 3 nutrients-17-00538-f003:**
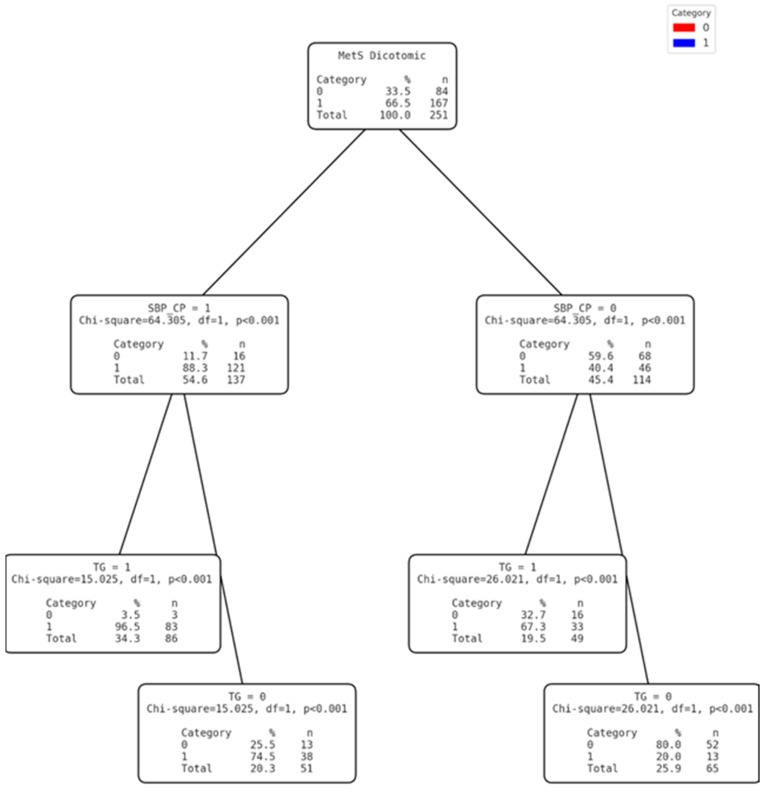
Clinical decision tree for MetS based on SBP CP(SBP Cut-off point) and TG.

**Figure 4 nutrients-17-00538-f004:**
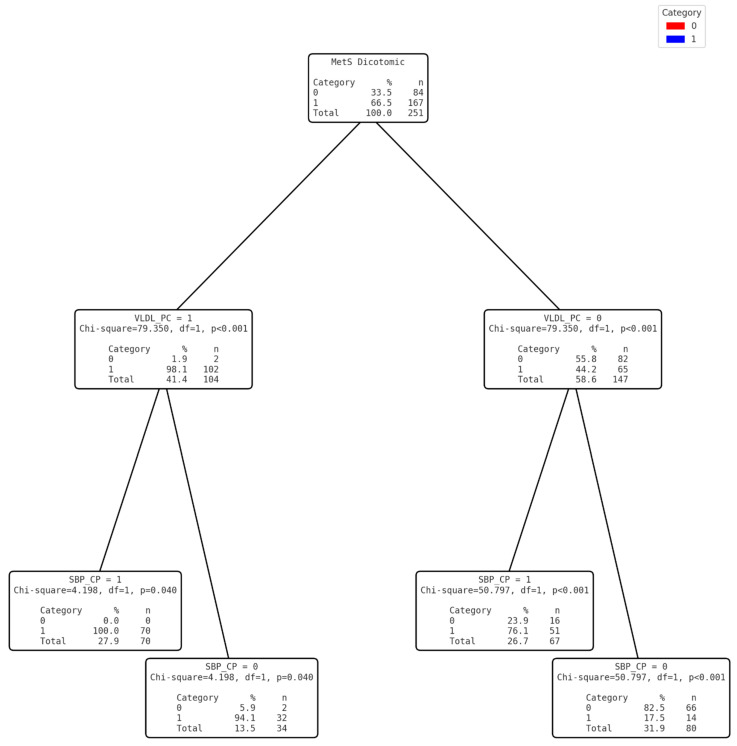
Clinical Decision Tree for MetS based on VLDL and SBP CP..

**Table 1 nutrients-17-00538-t001:** Sociodemographic and anthropometric characteristics.

Variable	Total *n* = 251	Women 164 (65.3%)	Men 87 (34.7%)	*p*
**Age (years)**	47.93 (SD 15.71)	47.68 (SD15.68)	48.39 (1SD3.96)	
**Anthropometry**				
**WC (cm)**	99.64 (SD9.65)	96.53 (53 (8.2)	105.49 (SD9.51)	<0.001
**Level of WC**				
**Low**	7 (2.8%)	3 (42.9%)	4 (57.1%)	
**High**	244 (97.2%)	161 (66%)	83 (34%)	NS
**Weight (kg)**	73.83 (SD15.17)	67.98 (SD11.92)	71.57(SD6.48)	<0.001
**Height (cm)**	154.76 (SD1.54)	1.51 (SD0.05)	1.62 (SD0.06)	<0.001
**WHtR**	0.64 (SD0.07)	0.63 (SD0.07)	0.65 (SD0.05)	
**Level of WHtR**				
**Low**	5 (2)	4 (80)	1 (20%)	NS
**High**	244 (98)	158 (64.8)	86 (35.2%)	
**BMI (kg/m^2^)**	30.360 (SD4.86)	29.83(SD 3.12)	32 (SD4.44)	<0.001
**Level of BMI**				
**Underweight**	2 (0.8%)	2 (100%)	0	<0.001
**Healthy weight**	23.8 (9.2%)	20 (87%)	3 (13%)	
**Overweight**	96 (38.2%)	69 (71.1%)	27 (28.1%)	
**Obesity**	119 (47.4%)	66 (55.5%)	52 (44.5%)	

Quantitative variables with mean and SD. Qualitative variables with absolute frequency and percentage; BMI: Body Mass Index; WC: Waist Circumference; WHtR: waist-to-height ratio. SD: Standar Desviation; NS: No Significant

**Table 2 nutrients-17-00538-t002:** Characteristics according to bioimpedance and clinical variables.

Variable	Total	Women 164 (65.3%)	Men 87 (34.7%)	*p*
**Bioimpedance variables**				
**FP (%)**	32.73 (SD7.17)	35.21 (SD6.67)	28.06 (SD5.61)	<0.001
**MM (kg)**	46.86 (SD 4.05)	47.26 (SD5.53)	52.39 (SD4.27)	<0.001
**BMR (kcal)**	1492.53 (SD296.48)	1334.61 (SD162.31)	1791.02 (SD260.85)	<0.001
**MA (years)**	51 (SD10.91)	49 (SD10.8)	54.8 (SD10.13)	<0.001
**Clinical variables**				
**SBP (mmHg)**	130.37 (DE 23.65)	125.41 (SD 23.44)	139.70 (SD 21.21)	<0.001
**DBP (mmHg)**	79.06 (DE 13.07)	75.68 (SD 12.83)	85.43 (SD 11.04)	<0.001
**HDL-C: (mg/dL)**	38.18 (DE 13.31)	42.38 (SD 12.61)	30.24 (SD 6.479)	<0.001
**Level of HDL-C**				
**Low**	49 (19.5%)	39 (79.6%)	10 (20.4%)	<0.05
**High**	202 (80.5%)	125 (61.9%)	77 (38.1%)	
**TG (mg/dL)**	152.12 (SD 69.5)	134.01 (SD 124.01)	170.44 (SD 76.17)	<0.001
**TG level**				
**Low**	147 (58.6%)	106 (72.1%)	41 (27.9%)	<0.05
**High**	104 (41.4%)	58 (55.8%)	46 (44.2%)	
**FG (mg/dL)**	103.42 (SD 41.79)	102.63 (SD 39.42)	104.91 (SD 46.13)	NS
**Low**	169 (67.3%)	112 (66.3%)	57 (33.7%)	NS
**High**	82 (32.7%)	52 (63.4%)	30 (36.6%)	
**VLDL**	70.10 (SD 37.81)	45.48(SD 14.29)	82.45 (SD 39.87)	<0.001

Quantitative variables with mean and SD. Qualitative variables with absolute frequency and percentage; FP: fat percentage; MM: muscle mass; BMR: basal metabolism rate; MA: metabolic age; SST: systolic blood pressure; DBP: diastolic blood pressure; HDL-C: cholesterol; TG: triglycerides; FG: fasting glucose; VLDL: very low-density lipoprotein. SD: Standar Desviation; NS: No Significant.

**Table 4 nutrients-17-00538-t004:** Comparison of diagnostic accuracy between variables.

Variable	AUC	*p*	95%IC	Cut-Off Point	Sensitivity	Specificity	J
**TG**	0.83	<0.001	0.78 –0.88	149.5	0.611	0.976	0.587
**VLDL**	0.82	<0.001	0.77–0.872	68.89	0.591	0.914	0.612
**SBP**	0.819	<0.001	0.71–0.83	125.5	0.725	0.81	0.534
**DBP**	0.756	<0.001	0.69–0.82	75.5	0.737	0.69	0.424

TG: Triglycerides; SBP: Systolic Blood Pressure; DBP: Diastolic Blood Pressure; VLDL: Very Low-Density Lipoprotein; AUC: Area Under the Curve; IC: Confidence Interval; J: Youden Index.

**Table 5 nutrients-17-00538-t005:** Comparative diagnostic accuracy of clinical decision trees.

	Sensitivity (%)	Specificity (%)	Validity Index (%)	PPV (%)	NVP (%)	Youden
**Model 1**	84.43 (78.63–90.23)	65.48 (54.71–76.24)	78.09 (72.77–83.40)	83.94 (76.99–88.89)	67.90 (57.12–79.69)	0.5 (0.38–0.61)
**Model 2**	92.22 (87.85–96.58)	61.90 (50.92–72.88)	82.07 (77.13–87.02)	82.80 (77.10–88.49)	80 (69.51–90.49)	0.54 (0.43–0.65)
**Model 3**	91.62 (87.11–96.12)	78.57 (69.2–87.94)	87.25 (82.93–91.58)	89.47(84.58–94.37)	82.50(73.55–91.45)	0.7(0.6–0.8)

## Data Availability

The data are not publicly available due to privacy or ethical restrictions.

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
