# Peer review of "Metabolic Syndrome in the Amazon: Customizing Diagnostic Methods for Urban Communities"

_nutrients, 2025, doi:10.3390/nu17030538_

Round 1
Reviewer 1 Report
Comments and Suggestions for Authors
Dear authors, I read the article intitulated "Metabolic Syndrome in the Amazon: Customizing Diagnostic Methods for Urban Communities". The subject is very interesting and show the reality of the city and the necessity of the new health policies. The results are obtain in some city of Peru. It is necessary to have more participants than 251 to discuss about the risk of MS. We need more informations about number of calories, activity, stress management to establish if the risk of MS development is so big.
Reviewer 2 Report
Comments and Suggestions for Authors
Abstract
• The abstract does not address potential limitations of the model, such as the need for external validation or the limited representativeness of the sample.
• There is no mention of the study’s limitations in terms of generalizability.
Introduction
• The relationship between urbanization and metabolic syndrome is well described, but there is a lack of discussion about the challenges and limitations of adapting international models to local contexts.
• A deeper engagement with prior studies on the validity of similar diagnostic models in comparable populations is missing.
Materials and Methods
• The use of a non-probabilistic convenience sample limits the generalizability of findings and introduces selection bias.
• The reliance on traditional anthropometric and biochemical variables does not explore advanced biomarkers or digital health solutions that could improve diagnostic accuracy.
• The cross-sectional design prevents causal inferences and does not allow for an assessment of the model’s long-term predictive power.
• The choice of the Youden index for determining cut-off values might limit clinical applicability, as it does not always align with real-world conditions.
Results
• The relatively small sample size (251 participants) reduces statistical power and generalizability.
• Validation was limited to the study population, raising questions about applicability to other demographic or geographic groups.
• The analysis lacks detail on subgroup differences, such as those based on gender or age.
Discussion
• The limitations of the study, particularly regarding generalizability and the need for further validation, are only briefly mentioned.
• The comparison with similar models is superficial and does not highlight the advantages or disadvantages of the proposed approach.
• There is no discussion of the potential integration of digital technologies or innovative approaches to improve the model.
• The discussion does not address the long-term effectiveness of the model through follow-up or longitudinal studies.
Conclusions
• The conclusions are overly optimistic and emphasize the model’s efficiency without adequately addressing methodological limitations.
• Recommendations for future research or practical implementation are not clearly outlined, such as expanding the range of biomarkers or incorporating other technologies.
Ethical Considerations
• While the study adheres to ethical standards, there is no detailed explanation of how cultural and social characteristics of the population were addressed to ensure a representative sample.
Reviewer 3 Report
Comments and Suggestions for Authors
The manuscript entitled Metabolic Syndrome in the Amazon: Customizing Diagnostic Methods for Urban Communities is an original article. The authors aimed to develop and validate a simple, cost-effective diagnostic model for early detection of metabolic syndrome in the urban population of San Juan Bautista, Iquitos. They identified an effective diagnostic model with a high diagnostic accuracy. This research project was a Doctoral Thesis.
There are some issues related to this study.
First of all, it is a small study. The results can be apllied only in this population study.
I wondering how reliable are bioimpedance variables for a research study?
In table 1 : What is Nivel TG ? Do you mean TG level ? And why low and high TG level? Why was not sufficient mean value of TG (and abnormal level that means high level)? When was measured the values of TG in all population included in this study? All biological parameters was assessed a-jeune?
Why did you nou used Triglyceride and Glucose Index in this study ?
Heart rate is not important to be included in Table 1 ?
How about alcohol consommation which can influence the level of TG? Do you have data about this biological parameter in this urban population of San Juan Bautista? Please add in discussions.
I recommend keeping only table 2, because there are data which are repeated in table 1.
It seems that there are many acronyms, some of those for only ones. For example: Nutritional Transition (NT). Please verify this issue.
Round 2
Reviewer 2 Report
Comments and Suggestions for Authors
None